# Neuroendocrine Carcinomas of the Digestive Tract: What Is New?

**DOI:** 10.3390/cancers13153766

**Published:** 2021-07-27

**Authors:** Anna Pellat, Anne Ségolène Cottereau, Benoit Terris, Romain Coriat

**Affiliations:** 1Gastroenterology and Digestive Oncology Unit, Cochin Teaching Hospital, AP-HP, Université de Paris, 27 rue du Faubourg Saint Jacques, 75014 Paris, France; romain.coriat@aphp.fr; 2Nuclear Medicine Department, Cochin Teaching Hospital, AP-HP, Université de Paris, 27 rue du Faubourg Saint Jacques, 75014 Paris, France; annesegolene.cottereau@aphp.fr; 3Pathology Department, Cochin Teaching Hospital, AP-HP, Université de Paris, 27 rue du Faubourg Saint Jacques, 75014 Paris, France; benoit.terris@aphp.fr

**Keywords:** neuroendocrine carcinomas, small-cell lung cancer, digestive tract, chemotherapy, peptide receptor radionuclide therapy, immunotherapy

## Abstract

**Simple Summary:**

In this narrative review, we describe the current data and management of neuroendocrine carcinomas (NEC) of the digestive tract. These tumors are very rare and suffer from a lack of clinical trials which would allow for standardized therapeutic management. To date, most guidelines come from studies in small-cell lung cancer, which is a similar entity in the lung. The incidence of NEC is rising and their prognostic is very low, underlying the urgent need for more trials to help define their best management.

**Abstract:**

Neuroendocrine carcinomas (NEC) are rare tumors with a rising incidence. They show poorly differentiated morphology with a high proliferation rate (Ki-67 index). They frequently arise in the lung (small and large-cell lung cancer) but rarely from the gastrointestinal tract. Due to their rarity, very little is known about digestive NEC and few studies have been conducted. Therefore, most of therapeutic recommendations are issued from work on small-cell lung cancers (SCLC). Recent improvement in pathology and imaging has allowed for better detection and classification of high-grade NEN. The 2019 World Health Organization (WHO) classification has described a new entity of well-differentiated grade 3 neuroendocrine tumors (NET G-3), with better prognosis, that should be managed separately from NEC. NEC are aggressive neoplasms often diagnosed at a metastatic state. In the localized setting, surgery can be performed in selected patients followed by adjuvant platinum-based chemotherapy. Concurrent chemoradiotherapy is also an option for NEC of the lung, rectum, and esophagus. In metastatic NEC, chemotherapy is administered with a classic combination of platinum salts and etoposide in the first-line setting. Peptide receptor radionuclide therapy (PRRT) has shown positive results in high-grade NEN populations and immunotherapy trials are still ongoing. Available therapies have improved the overall survival of NEC but there is still an urgent need for improvement. This narrative review sums up the current data on digestive NEC while exploring future directions for their management.

## 1. Introduction

Neuroendocrine neoplasms (NEN) are defined by the expression of specific diagnostic tissue biomarkers, such as synaptophysin and chromogranin A (CGA). These tissue biomarkers are very sensitive and specific in the well-differentiated setting but can be lacking in high-grade NEN [1,2,3]. The 2019 World Health Organization (WHO) classification differentiates poorly differentiated digestive NEN from well-differentiated NEN (Table 1) [4]. Neuroendocrine carcinomas (NEC) show poorly differentiated morphology with a high proliferation rate (Ki−67 index > 20%), and can be divided according to cell size (small-cell or large-cell). This 2019 classification has introduced a new category of tumors named well-differentiated grade 3 neuroendocrine tumors (NET G−3) officially separating them from NEC [2,5,6,7]. This has been a major change in the field, formalizing that high-grade NEN are a heterogenous population and that various G-3 entities should be considered separately in ongoing and future clinical trials. Indeed, NET G-3 have a better prognosis than NEC due to different tumor characteristics [7,8,9,10] so current available data on high-grade NEN is not up to date with an urgent need for prospective studies focusing on these different entities. Diagnosis and management of NET G-3 is detailed in another paper of this special issue. 

On top of that, most of guidelines on NEC management derive from trials in small-cell lung cancer (SCLC), which is the closest tumoral entity in the lung. This is mainly due to the rarity of digestive NEC and the lack of specific clinical trials. Improvement in pathology diagnosis, functional imaging, and treatment, has helped individualize them and increase interest for their specialized management.

We present in this narrative review the latest data on clinical, radiological, and histopathological presentation, as well as treatment, of NEC of the digestive tract.

## 2. Epidemiological Features and Tumor Presentation

### 2.1. Incidence and Tumor Site

Digestive NEN are rare tumors but with a growing worldwide incidence due to better identification [11,12,13]. Poorly differentiated NEN are an even rarer subgroup with proportions that vary greatly between studies, from about 9% to 75% of all NEN, depending on the consideration of SCLC in the final analysis [11,12,14]. In Europe, the proportion of digestive NEC varies from 3.4% to 30.3%, depending on the country [15,16]. Small series focusing on high-grade NEN have found that NEC represent about 69% to 80% of this population [14,17,18]. More recently, the American Surveillance, Epidemiology, and End Results (SEER) database showed that out of 162 983 patients diagnosed with poorly differentiated NEN between 1973 and 2012, 37.4% were of gastrointestinal location with an incidence of 0.04 per 100,000 [19]. The highest incidence was found in the respiratory system, with 8.4 per 100,000 [19]. 

Regarding tumor site, poorly differentiated NEN are most frequently found in the lung [12,19]. In the SEER registry 148 251 of lesions (91.3%) were diagnosed in the respiratory system [19]. NEC can also be found anywhere along the digestive tract, but most frequent tumor sites are the colon and rectum (from 27 to 38%), the pancreas (about 20%), and the esophagus (11 to 22%) [19,20,21]. Therefore, most of the current data comes from gastro–entero–pancreatic NEC (GEP-NEC) series. Tumor site incidence also varies according to sex and cell size [19,20]. Indeed, most NEC found in the colon, rectum or pancreas are large-cell tumors, whereas most NEC found in the esophagus are small-cell tumors [19]. 

### 2.2. Clinical Presentation and Biomarkers

Digestive NEC are rarely part of genetic syndromes and almost always sporadic. To date, there are no clear risk factors identified for NEC. 

Clinical symptoms essentially depend on initial tumor location as well as the metastatic status. Digestive NEC are often aggressive tumors with systemic symptoms [22]. Metastases are often found in the liver or peritoneum rather than the lung or brain [22,23,24]. NEC are more frequently diagnosed in men, irrespective of initial tumor site [19]. Functioning syndromes are extremely rare in NEC compared with well-differentiated NET [6,18,22,25] but Cushing syndromes have been described [26]. 

In the absence of a carcinoid syndrome, there is no use for urinary 5-HIAA monitoring in NEC patients [27]. Other specific biomarkers, such as plasma CGA and plasma neuron specific enolase (NSE), are often dosed for NEC management even if there is little data in this population. Assessment of NSE is recommended in SCLC where it mainly has a prognostic value, but can also help monitor treatment evolution [28]. In GEP-NEN, one study found that poorly differentiated NEN more frequently presented with elevated NSE compared with well-differentiated tumors: 12/19 (63%) versus (vs.) 23/71 (32%), *p* = 0.01 [29]. Some works also suggested that NSE had a negative prognostic value on survival for NEC patients when elevated at baseline [21,30]. Finally, plasma CGA can also be elevated in NEC but with no significant differences in concentration levels compared with well-differentiated NET [21,30].

Overall, there is no unique clinical presentation for NEC patients. Additionally, data is scarce regarding plasma biomarkers with only small populations evaluated. We propose the dosage of both plasma CGA and NSE at diagnosis, which should only be repeated if elevated at baseline, to help the clinician with therapeutic decision. 

### 2.3. Tumoral Staging and Prognosis

Both tumor differentiation and grade, as represented in the WHO classification, are important prognostic markers in NEN. Poorly differentiated high-grade NEN, or NEC, have a worse prognosis than other subtypes. In 2013, a study in the Netherlands on 47,800 patients with NEN from all sites found that survival seemed strongly correlated with grade: 5-year overall survival (OS) rates were 80%, 63%, 20%, and 6%, for G-1, G-2, large-cell G-3, and small-cell G-3 NEN, respectively (no *p*-value) [12]. Regarding differentiation, small series comparing NET G-3 with NEC have shown that NET G-3 have a better prognosis than NEC [5,6,18,31,32].

Cell size also seems to influence survival in NEC. In the SEER database, significant differences were found according to morphological subtype (*p* < 0.001), with small-cell histology being associated with worse median and 5-year survivals at most digestive sites [19].

Tumor stage at diagnosis also influences survival [33]. The majority of patients with NEC are diagnosed at a metastatic stage, with values ranging from 60% to 78% in different works [12,19,21,24]. Median OS for metastatic gastrointestinal NEC is about 5.2 months vs. 33.9 months in the localized setting [11,19]. The 5-year survival rate is about 4.7% in metastatic GEP-NEC vs. 42% in the localized setting [19]. These values vary with tumor site, with better survivals observed in the small intestine, colon, and rectum, compared with other digestive locations or the lung [19]. Although prognosis is worse in metastatic tumors, OS in NEC is low due to frequent recurrence of the disease even in the localized setting [23,26]. Survival is increased with chemotherapy administration: median survival of metastatic NEC without chemotherapy is one month (IC 95% 0.3–1.8) vs. 11 to 19 months with chemotherapy (IC95% 9.4–12.6) (see Treatment section) [18,21,25,34,35,36]

Additionally, several factors such as poor performance status, high tumor burden, presence of liver metastases, high Ki-67 index and elevated lactate dehydrogenase (LDH), NSE and CGA at baseline can negatively influence survival in patients with metastatic NEC [21,25,32,37,38]. A gastrointestinal NEC (GI-NEC) score, based on several of these previous markers, has been developed to help with therapeutic decisions in the GEP-NEC population: it identified two different subgroups with different prognosis, suggesting it could be used as a stratification marker in future trials [38]. 

To sum up, digestive NEC are aggressive non-functional tumors often diagnosed at a metastatic stage. They represent the highest proportion of NEN G-3 (up to 80%). A high-grade functional NEN, especially in the presence of cancer predisposition, should favor the diagnosis of NET G-3 rather than NEC.

## 3. Imaging

Imaging plays a crucial role in NEC for diagnosis, initial staging as well as evaluation of treatment response. Functional imaging, along with peptide receptor radionuclide therapy (PRRT), has clearly changed NEN treatment approach.

### 3.1. Morphological Imaging

Morphological imaging, including thoraco-abdominopelvic computed tomography (TAP-CT) and liver magnetic resonance imaging (liver MRI), is fundamental to characterize initial tumor stage and to monitor treatment response [39]. These two types of imaging have the advantage of being widely available and reproductible. MRI is preferred for examination of the liver, the pancreas, and the brain [40]. Regarding diagnosis, several morphologic features such as ill-defined margins, large tumor size, heterogeneous and poor-to-moderate enhancement, vascular involvement, and upstream Wirsung duct dilatation are more frequently observed in pancreatic NEN G-3 (PanNEN G-3) than PanNET G-1/G-2 [41,42,43]. In MRI, values of apparent diffusion coefficients are significantly lower for PanNEC compared with other PanNEN [41]. In a recent work, 23 patients with NEC presented with larger sized tumors, more necrosis, and lower attenuation on pre-contrast and on portal venous phase CT images, with all results being significant compared with NET G-3 [44]. Hemorrhagic content on MRI was only observed in NEC (*p* = 0.007) [44]. Finally, it has also been shown that different subgroups of PanNEN have a suggestive CT radiomics signature that helps differentiate them according to their grade [42,43,45]. 

Brain MRI is recommended at baseline in case of diagnosis of a large or small-cell NEC of the lung, to look for brain metastases. In digestive NEC, brain metastases are less frequently encountered with values ranging from 0 to 4% at diagnosis in different series [22,23,24,25]. Although the European Neuroendocrine Tumor Society (ENETS) guidelines recommend performing a brain MRI in digestive NEC at diagnosis, these recommendations are not consensual and some only recommend it in the presence of neurological symptoms [46].

Regarding overall tumor staging, historical series have often classified digestive NEC as localized or extensive, as for NEC of the lung. To date, the ENETS guidelines recommend that for all NEC the international TNM staging system of adenocarcinomas must be applied.

### 3.2. Functional Imaging

18F-Fluorodesoxyglucose (18F-FDG) positron emission tomography (PET)-CT is recommended to help for initial NEC diagnosis and staging (Figure 1). FDG is the tracer of choice for aggressive tumors with higher glucose metabolism and less somatostatin receptors (SSTRs) expression. Indeed, it has been shown that there is a correlation between high Ki-67 index values and positivity of 18F-FDG PET-CT in NEN [47,48]. In a recent series of 119 patients with GEP NEC, 110 patients had a positive 18F-FDG PET-CT [21]. Furthermore, positive 18F-FDG PET-CT is associated with poor prognosis and survival in NET [49]. 

Somatostatin receptor imaging (SRI), such as ^68^Ga-DOTA-somatosatin analogue (SSA) PET-CT, can also be individually discussed and performed in NEC patients for optimal diagnostic and prognostic information. Indeed, dual tracer can help classify and select some patients for individualized treatment with PRRT, which has shown promising results in high-grade NEN (see Treatment paragraph) [50]. PRRT is feasible for lesions with both SSTRs overexpression and high glucose uptake, with little mismatch. By contrast, a predominant FDG-avid disease showing low or absent SSTRs expression is commonly considered as an exclusion criterion for PRRT eligibility, which is often the case in NEC.

To sum up, TAP-CT and 18F-FDG PET-CT should both be performed at baseline after NEC diagnosis and can be completed with liver and brain MRI, depending on tumor presentation. Dual tracer can be proposed on an individual basis to help for therapeutic decision, keeping in mind that a positive SRI uptake in high-grade NEN is usually more in favor of NET G-3 rather than NEC.

## 4. Histology

### 4.1. Morphological and Immunohistochemistry Features

Pathological evaluation in high-grade NEN can sometimes be challenging. In France, it is recommended to have a second evaluation by an expert NEN pathologist from the TENPATH group (expert pathologists’ network) for every new case of high-grade NEN, including NEC. Pathological study of series of high-grade PanNEN have shown that there are morphological differences between PanNET G-3 and PanNEC. Indeed, in PanNEC tumor cells show less plasmacytoid morphology, frequent lack of abundant cytoplasm, frequent molding and nuclear tangles, and fibrosis [51,52]. Evidence of high proliferation in NEC can be seen through changes in morphology such as apoptosis and mitoses, as well as a high proportion of necrosis [52]. The presence of another histological type can also result in tumor diagnosis difficulty (mixed morphology). In 2017, the WHO classification has introduced the notion of MiNEN where any other histological type can be associated with the neuroendocrine morphology (at least 30% of the tumor sample) (Table 1) [8].

As previously mentioned, NEN express positive labeling for neuroendocrine tissue markers such as synaptophysin and CGA. CGA labeling is often lacking in NEC whereas synaptophysin is frequently expressed [25,33]. High-grade NEN expressing both synaptophysin and CGA seem to have a better prognosis [53,54]. 

As for NEC of the lung, digestive NEC show small or large-cell morphology but sometimes the distinction between these 2 entities is not obvious, even for expert pathologists. Gastrointestinal NEC are more frequently large-cell tumors, except for a few tumor sites such as the esophagus, the gall bladder, and the anal canal [19]. Current management does not differ between these two entities, but previous work suggested differences in tumor presentation as well as prognosis [19]. 

Regarding the proliferation rate, NEC often show high Ki-67 index values, up to 100%, compared with NET G-3 [3,5,6,18,31,51]. Therefore, an accurate pathological assessment of the Ki-67 proliferation index is essential for NEN diagnosis. Various technical factors may potentially affect its reproducibility (specimen type, choice of antibody and the assessment method) [55]. Manual counting (MC) of >2000 cells is considered the “gold standard” method by the WHO grading system and can be performed through the microscope or on screenshot printed image, which seems the most practical method based on its cost/benefit ratio and reproducibility [56].

We here present the pathological features of a NEC of the colon (Figure 2).

### 4.2. Molecular Biology

Morphological characterization of high-grade NEN can be difficult when there is important tumoral heterogeneity and/or necrosis [5]. Following the results of the NORDIC study, some authors have suggested that the 55% Ki-67 value could be the best cutoff to distinguish well-differentiated NEN G-3 from NEC [25,57]. To this day, this has not been validated and molecular biology data are currently used to help for the distinction between NEC and NET G-3. 

Some researchers have suggested that PanNEC could develop from ductal adenocarcinoma with common key genetic drivers [52,58]. Indeed, digestive NEC frequently carry mutations of adenocarcinoma, such as *KRAS* and *BRAF* [59,60]. Additionally, gastrointestinal NEC show frequent inactivation of the TP53, Rb, and SMAD4 pathways, due to intragenic mutations in the *TP53*, *RB1,* and *SMAD4* genes [61,62,63,64]. These genetic changes are rarely seen in well-differentiated NEN [61,64]. Regarding NEC cell subtype, no difference was seen in *RB1* and *KRAS* mutations [3]. Finally, high expression of p16 and BCL-2 was also seen in colorectal NEC [58]. Overall, NEC seem to be a heterogeneous population which could be divided according to main molecular signatures which could influence treatment response: “small-cell” or “adenocarcinoma” type, for instance [3,58,65]. This needs confirmation in future studies.

### 4.3. Circulating Tumor Deoxyribonucleic Acid (DNA)

Circulating tumor deoxyribonucleic acid (ctDNA) analysis, or liquid biopsy, has become routine practice for many malignancies. It can be highly informative but suffers from various technique difficulties [66]. There is little data regarding ctDNA in NEC management. Recently, a pilot study has shown that for 24 patients with NEC, 22 had at least one driver mutation [67]. Tumors showed heterogeneous alterations, with the exception of the *TP53* mutation which was present in 83% of cases. There was a 44% agreement between ctDNA and tissue NGS alterations [67]. Data is lacking for ctDNA in resected NEC due to the rarity of this situation. The NEONEC study (Eudra CT 2019-004096-39), exploring perioperative chemotherapy in localized digestive NEC, will investigate ctDNA in this situation and hopefully bring some answers. Further studies should investigate the role of ctDNA in NEC management.

## 5. Treatment 

NEC therapeutic management suffers from a lack of well conducted clinical trials. Many recommendations derive from data in SCLC studies. Trials with new therapies are currently ongoing in the metastatic setting.

### 5.1. In the Localized Setting

#### 5.1.1. Surgery

Surgery is still controversial in localized digestive NEC. Available results mainly come from retrospective series where information on tumor differentiation is often lacking. For NEC of the colon and rectum data are contradictory. One work has shown no benefit of surgery of the primary tumor on survival, in both localized and metastatic colorectal NEC [26], whereas another suggested significant higher survival for surgery of localized large-cell colorectal NEC [68]. Nevertheless, in Smith et al. work, some operated patients had no evidence of disease recurrence after a median follow-up of 37 months [26]. Some case reports and small series have also shown long survivals for patient operated from a NEC of the esophagus [69,70]. Retrospective series of small series of operated PanNEC found median survival times ranging between 11 and 23 months after surgery [71,72]. Finally, a recent study found that OS was longer after surgery of localized PanNEC vs. no surgery, but without significant difference (*p* = 0.093). One single-center retrospective study evaluated survival in patients who were endoscopically or surgically treated for gastric NEN. For the 69 patients with gastric NEN G-3 (no information on differentiation), including one metastatic patient, median OS was 19 months, which was significantly lower than for NEN G-1 patients (*p* < 0.001) [73]. To our knowledge, there is no specific data or large series regarding surgery in midgut NEC or other digestive sites. Overall, surgery of localized NEC should only be proposed for highly selected patients, and preferably for colorectal NEC.

Various retrospective series have suggested significant higher survival with adjuvant chemotherapy [72,74,75] but there is currently no available prospective data. Neoadjuvant chemotherapy has been evaluated in a few patients without enough data to conclude [26,68,75]. The most frequently evaluated chemotherapy regimen was a combination of platinum-based and etoposide molecules. The ongoing French NEONEC study (Eudra CT 2019-004096-39) will evaluate the impact of perioperative chemotherapy in localized digestive NEC. Overall, based on these preliminary results and therapeutic guidelines in SCLC, adjuvant chemotherapy with four cycles of platinum-based chemotherapy is recommended in digestive NEC after surgery.

#### 5.1.2. Radiotherapy and Concurrent Chemoradiotherapy

In localized SCLC, radiotherapy combined with platinum-based chemotherapy improved survival and is currently recommended in this setting [76,77]. There is very little data on the role of radiotherapy in localized digestive NEC. One work on 14 patients with localized small-cell NEC of the esophagus found a median survival of 22.3 months, with six patients treated by chemotherapy followed by chemoradiotherapy [78]. A more recent study in patients with NEC of the anal canal found similar OS with chemoradiotherapy compared with surgery (OS of 49.1 months vs. 39.2 months, *p* = 0.42) [79]. Chemoradiotherapy can be individually discussed and proposed in the localized setting for patients with NEC of the esophagus or the anal canal, especially when surgery is not recommended or at risk.

Prophylactic brain irradiation is not recommended in digestive NEC where brain metastases are rarely encountered. 

### 5.2. In the Metastatic Setting

#### 5.2.1. Surgery

One study has suggested higher survival when operating 12 patients with metastatic PanNEC [72] whereas others showed no improvement on survival [26,80]. To date, surgery is not recommended in metastatic digestive NEC.

#### 5.2.2. Liver-Directed Therapies

Interventional radiology plays an important role in the therapeutic management of well-differentiated low-grade NEN, especially when liver burden is important or in the presence of a secretory syndrome [81,82,83]. Various types of liver-directed therapies are currently available, including intra-arterial therapies such as transarterial embolization (TAE), transarterial chemoembolization (TACE) and selective internal radiation therapy (SIRT), or radiofrequency ablation (RFA). In a small series of metastatic GEP-NEN G-3, one retrospective study suggested that aggressive locoregional treatment, including treatment with intra-arterial liver-directed therapies, improved OS compared with administration of systemic chemotherapy alone [84]. To date, there is not enough data to recommend this therapeutic approach in routine practice for NEC.

#### 5.2.3. First-Line Chemotherapy 

Chemotherapy is the main available treatment in metastatic digestive NEC and should be urgently administered after diagnosis [40]. Again here, the majority of data comes from retrospective or phase II studies, and prospective clinical trials are warranted. The available trials are heterogeneous regarding included population (mixed tumor sites), tumor histology (e.g., differentiation, presence of MiNEN) as well as the chemotherapy regimen used (Table 2), so results need to be handled with caution. With chemotherapy, response rates (RR) and median OS ranged from 14 to 67% and from 5.8 to 19 months respectively. The two most frequently used chemotherapy regimen were platinum-based with either irinotecan or etoposide. Indeed, following Nakano et al. work, the combination of cisplatin and etoposide has been approved as first-line treatment [85]. In clinical practice, and following current guidelines in SCLC, six cycles of chemotherapy should be administered in metastatic digestive NEC. In the absence of progressive disease, surveillance can be started. An ongoing trial, FOLFIRINEC (NCT04325425), will evaluate the effect of mFOLFIRINOX (combination of irinotecan, oxaliplatin and 5FU) vs. platinum-based chemotherapy in the first-line setting for NEC of the digestive system and of unknown primary. 

#### 5.2.4. Second-Line Chemotherapy 

There is no standard of care chemotherapy in the second-line setting for metastatic digestive NEC. Combinations of 5FU and irinotecan (FOLFIRI) and 5FU and oxaliplatin (FOLFOX or XELOX) can be proposed based on results from small retrospective studies (Table 3). A few studies have also found promising results with temozolomide-based chemotherapy, especially when Ki-67 was <55% [25,54,91,92]. As mentioned previously, historical series have included heterogeneous populations of well and poorly differentiated NEN G-3, so results are also to handle with caution here. The MGMT (6-O methylguanine-ADN methyltransferase) status should be determined before administration of temozolomide-based chemotherapy. 

#### 5.2.5. Peptide Receptor Radionuclide Therapy (PRRT)

SSTR may also be targeted with radiolabeled SSA such as 177Lu-DOTA-D-Phe-Tyr3-octreotate (177Lu-DOTATATE) for PRRT. PRRT has been evaluated in the second or third-line setting for high-grade NEN patients showing foci anatomical agreement of the SRI and glucose uptake lesions (little mismatch). A recent review of four retrospective studies, which included GEP-NEN G-3 patients treated with PRRT, showed promising RR (31–42%) and disease control rates (DCR) (69–78%) in this population [96]. Progression-free survival (PFS) was higher in NEC with Ki-67 between 21 and 55% compared with NEC with Ki-67 index >55% (11 vs. 14 months, *p* = 0.04). This was also true for OS (22 vs. 9 months, *p* = 0.009) [96]. Although the four considered studies showed differences in design, they all suggested that about two thirds of the pooled population of NEN G-3 had a potential to respond to PRRT. Recently, the combination of PRRT and temozolomide-based chemotherapy has been evaluated in advanced G-2 and G-3 NEN with dual tracer uptake, with results showing significant activity with mild toxicities [97].

In carefully selected patients, PRRT can be considered after first-line treatment for NEC with increased uptake on SRI and little mismatch, especially when Ki−67 index < 55% [40]. Here, dual tracer can provide important information for NEC patient selection for PRRT.

#### 5.2.6. Targeted Therapies

Targeted therapies such as sunitinib [98], a tyrosine kinase inhibitor, and everolimus [99], a mTOR inhibitor, have both demonstrated efficacy in randomized phase III trials for well-differentiated PanNET G-1 and G-2. Similarly, everolimus has shown some efficacy in non-pancreatic NET [100]. There is little data on the effect of targeted therapies in NEC. One work on 20 patients with NEC has shown evidence of sunitinib activity (partial responses and stabilizations) [101]. In another study on 15 patients with PanNEN G−3 tumors, administration of everolimus as first-line treatment showed sustained disease stabilization for three out of four patients [102]. Results of the EVINEC phase II trial, which evaluated everolimus as second-line treatment in NEN G-3, are not yet available (NTC02113800). To sum up, targeted therapies can be individually discussed for progressive pre-treated patients with metastatic NEC, but cannot be proposed as a standard of care.

#### 5.2.7. Immunotherapy

Immune checkpoint inhibitors (ICI) might be a promising treatment in high-grade NEN which more frequently show microsatellite instability, and/or high mutational load. Indeed, some case reports have reported treatment response or long survivals in this population treated with ICI [103]. Early results with programmed-death-1 blockage came from treatment in both first and second-line settings of Merkel cell carcinoma, a high-grade cutaneous NEC [104,105]. Despite these encouraging preliminary results, one study showed that pembrolizumab alone had limited effect in NEN G-3 patients (DCR rate of 24.1%) [106]. There are several ongoing phase II studies investigating the effect of ICI in patients with advanced high-grade NEN: avelumab in progressive NEC/NET G−3 after chemotherapy (NCT03352934), and the combination of durvalumab and tremelimumab in pre-treated GEP-NEN G3 (NCT03095274). Further studies should also investigate the combination of ICI and chemotherapy in NEC patients. 

## 6. Conclusions

Digestive NEC are rare tumors showing poorly differentiated morphology with high Ki-67 index values (>20%). They represent about 80% of the high-grade NEN population. They show heterogeneous clinical presentation, depending on tumor site, but are always aggressive tumors diagnosed at a metastatic stage in more than 60% of cases. NEC of the digestive tract should be managed in expert centers; one work showed that treatment at an academic center improved survival in this population [24]. Surgery followed by adjuvant chemotherapy, or chemoradiotherapy, can be proposed in the localized setting for highly selected patients. First-line platinum-based chemotherapy should be urgently administered in the metastatic setting. In second or third-line settings, other regimen of chemotherapy can be proposed and targeted therapies and PRRT can be individually discussed. Newly diagnosed NEC patients should be included in clinical trials as a priority in order to embellish the body of evidence for NEC therapeutic management.

## Figures and Tables

**Figure 1 cancers-13-03766-f001:**
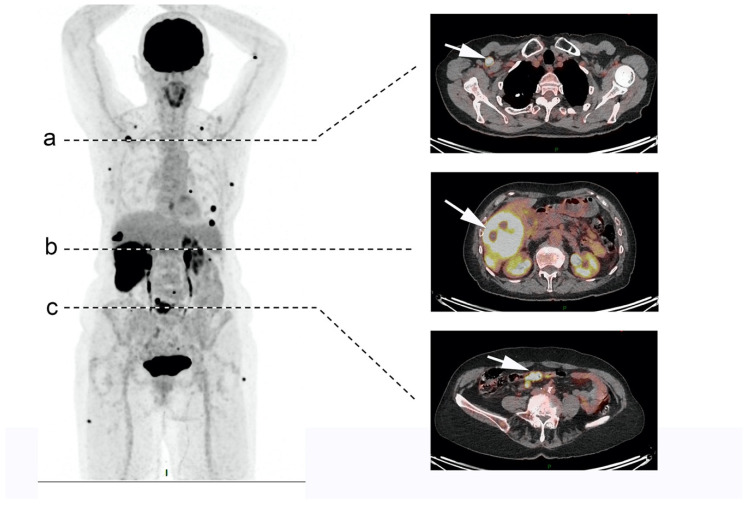
18F-Fluorodesoxyglucose (18F-FDG) positron emission tomography (PET)-CT results for a 79-year-old patient with metastatic neuroendocrine carcinoma (NEC) of the ileum (Ki-67 100%). Pre-therapeutic 18F-FDG PET/CT revealed disseminated metastatic disease with intense FDG uptake: multiple cutaneous lesions, (**a**) right infra clavicular node (SUVmax = 12), (**b**) large hepatic lesion (SUVmax = 21.6), and (**c**) focal primitive ileum lesion (SUVmax = 13.3).

**Figure 2 cancers-13-03766-f002:**
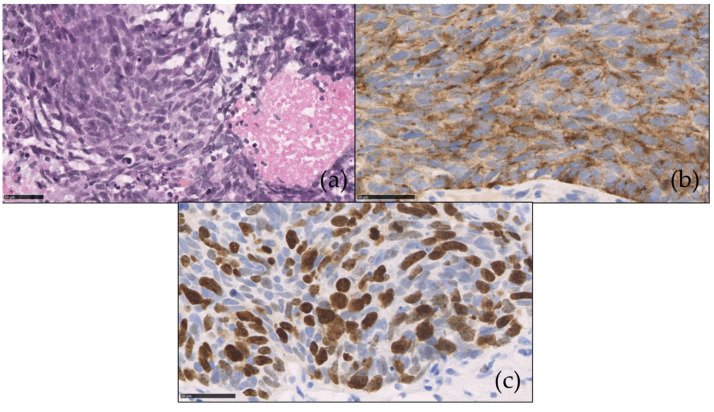
Poorly differentiated neuroendocrine carcinoma (NEC) of the colon (**a**) composed of large cells (Hematoxylin and eosin). Immunohistochemistry reveals (**b**) tumoral positivity for chromogranine A (**c**) with more than 70% of nuclei stained for Ki-67.

**Table 1 cancers-13-03766-t001:** The 2019 World Health Organization (WHO) classification for Neuroendocrine Neoplasms (NEN) of the digestive tract.

Well-Differentiated NEN ^1^	Ki−67 Index (%)	Mitotic Index (HPF ^2^)
NET ^3^ G-1 (low-grade)	<3	<2/10
NET G-2 (intermediate-grade)	3–20	2–20/10
NET G-3 (high-grade)	>20	>20/10
**Poorly differentiated NEN**		
NEC ^4^ G-3 Small-cell type, Large-cell type	>20	>20/10
Mixed Neuroendocrine–nonneuroendocrine neoplasm (MiNEN)		

^1^ NEN: neuroendocrine neoplasms, ^2^ HPF: high-power fields, ^3^ NET: neuroendocrine tumors, ^4^ NEC: neuroendocrine carcinomas.

**Table 2 cancers-13-03766-t002:** Results of studies evaluating first-line chemotherapy in metastatic neuroendocrine carcinomas (NEC).

Study (First Author, Year)	Tumor Site	Chemotherapy Regimen	Patients (Number)	Response Rate (%)	Median OS ^1^ (Months)
Moertel, 1991 [36]	Digestive Lung Unknown	Cisplatin/etoposide	18	67	19
Mitry, 1999 [34]	Digestive Lung Head and neck Uterus Unknown	Cisplatin/etoposide	41	41.5	15
Hainsworth, 2006 [86]	Digestive Lung Skin Thyroid Endometrium Prostate Sinus Unknown	Paclitaxel/carboplatin/ etoposide	78	42	14.5
Mani, 2008 [87]	NA	Cisplatin/irinotecan	20	58	NA ^2^
Iwasa, 2010 [35]	Digestive (hepatobiliary tract, pancreas)	Cisplatin/etoposide	21	14	5.8
Nakano, 2012 [85]	Digestive Urinary tract Head, neck Gynecologic Unknown	Cisplatin/irinotecan	44	50	16
Sorbye, 2013 [25]	Digestive	Cisplatin/etoposide	129	31	12
		Carboplatin/etoposide	67	30	11
		Carboplatin/etoposide/	28	44	10
		Vincristine			
		Other	28	NA	NA
Lu, 2013 [88]	Digestive	Cisplatin/irinotecan	16	51.1	10.6
Munhoz, 2013 [89]	Digestive Nasopharynx Prostate Unknown	Cisplatin or caboplatin/irinotecan	28	46.4	11.7
Yamaguchi, 2014 [37]	Digestive	Cisplatin/etoposide Cisplatin/irinotecan	46 160	28 50	7.3 13
Okuma, 2014 [90]	Oesophagus	Cisplatin/irinotecan	12	50	12.6
Walter, 2017 [21]	Digestive	Cisplatin or caboplatin/etoposide	152	50	11.6

^1^ OS: overall survival, ^2^ NA: non-available.

**Table 3 cancers-13-03766-t003:** Results of studies evaluating second-line chemotherapy in metastatic neuroendocrine carcinomas (NEC).

Study (First Author, Year)	Tumor Site	Chemotherapy Regimen	Patients (Number)	Response Rate (%)	Median OS ^1^ (Months)
Bajetta, 2006 [93]	Digestive Lung Other	Capecitabine/oxaliplatin (XELOX)	13	23	5
Welin, 2011 [54]	Digestive Lung Unknown	Temozolomide ± capecitabine ± bevacizumab	25	33	22
Olsen, 2012 [91]	Digestive Lung Prostate Kidney Unknown	Temozolomide	28	0	3.5
Hentic, 2012 [94]	Digestive	5-FU/irinotecan (FOLFIRI)	19	31	18
Sorbye, 2013 [25]	Digestive	Various combinations (temozolomide-based or taxotere-based)	100	18	19
Hadoux, 2015 [95]	Digestive Thoracic Other Unknown	5-FU/oxaliplatin (FOLFOX)	20	29	9.9
Walter, 2017 [21]	Digestive	5-FU/irinotecan (FOLFIRI)	72	24	5.9

^1^ OS: overall survival.

## Data Availability

This narrative review is based on previously published data.

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
