# Peer review of "Neuroendocrine Carcinomas of the Digestive Tract: What Is New?"

_cancers, 2021, doi:10.3390/cancers13153766_

Round 1

Reviewer 1 Report

This is a well designed review on all the aspects of NECs that have been published till recently. This review also underlines the need for prospective studies focusing on NECs with the currently used definition as staed in the introduction section, since in the most databases published the data of NECs and NETs G3 are presented together. A minor comment is to clearly state this latter comment in the introduction.

Author Response

We thank the reviewer for his suggestion. We have made the following changes in the revised version of the manuscript:

Modification in the revised manuscript, Introduction section, page 2:

“This has been a major change in the field, formalizing that high-grade (G-3) NEN are a heterogenous population and that various G-3 entities should be considered separately in ongoing and future clinical trials. Indeed, NET G-3 have a better prognosis than NEC due to different tumor characteristics [7–10] so current available data on high-grade NEN is not up to date with an urgent need for prospective studies focusing on these various entities. »

Reviewer 2 Report

The authors present a narrative review relating to current knowledge on neuroendocrine carcinoma of the digestive tract and suggest the need for further and more comprehensive clinical studies. The review is interesting and suggestive.

Author Response

We thank the reviewer for his comment about our review.

Reviewer 3 Report

The authors have performed a review on neuroendocrine carcinomas of the digestive tract. They have focused on new knowledge.

The review is nicely written. The presentation of the data is comprehensive.

My comments:

The authors stated that neuroendocrine neoplasms are defined by the expression of biomarkers such as synaptophysin and chromogranin. They should be more precise – synaptophysin is a tissue marker, chromogranin can be used as a tissue marker and as a marker in blood.

Abbreviations should be explained when mentioned for the first time. This is not always the case.

When talking about treatment (e.g. 5.1.1) the authors refer to colo-rectal NEC, oesophageal NEC and pancreatic NEC. Information on NEC located in the stomach or midgut is missing.

In the section Treatment in the metastatic setting (5.2.) information on liver-directed treatment options is lacking (e.g. RFA, TACE, SIRT).

Author Response

1/Comment n°1: The authors stated that neuroendocrine neoplasms are defined by the expression of biomarkers such as synaptophysin and chromogranin. They should be more precise – synaptophysin is a tissue marker, chromogranin can be used as a tissue marker and as a marker in blood.

Response

The reviewer is correct. We have made the following changes in the revised version of the manuscript:

Modification in the revised manuscript, Introduction section, page 2, line 44:

“Neuroendocrine neoplasms (NEN) are defined by the expression of specific diagnostic tissue biomarkers, such as synaptophysin and chromogranin A (CGA).”

Modification in the revised manuscript, Histology section, page 6, line 218:

“As previously mentioned, NEN express positive labeling for neuroendocrine tissue markers such as synaptophysin and CGA.”

2/Comment n°2: Abbreviations should be explained when mentioned for the first time. This is not always the case.

Response

The reviewer is correct. As examples, we have explained the following abbrevations:

Surveillance, Epidemiology and End Results (SEER)

Gastrointestinal NEC (GI-NEC)

Deoxyribonucleic acid (DNA)

3/Comment n°3: When talking about treatment (e.g. 5.1.1) the authors refer to colo-rectal NEC, oesophageal NEC and pancreatic NEC. Information on NEC located in the stomach or midgut is missing.

Response

We thank the reviewer for his comment. We have searched again the literature and found only one study on surgery in gastric NEC, but none in midgut NEC. We have made the following changes in the revised version of the manuscript:

Modification in the revised manuscript, Treatment section, page 7, line 286:

“One single-center retrospective study evaluated survival in patients who were endoscopically or surgically treated for gastric NEN. For the 69 patients with NEN G-3 (no information on differentiation), including one metastatic patient, median OS was 19 months, which was significantly lower than for NEN G-1 patients (p<0.001) [73]. To our knowledge, there is no specific data or large series regarding surgery in midgut NEC or other digestive sites. Overall, surgery of localized NEC should only be proposed for highly selected patients, and preferably for colorectal NEC.”

4/Comment n°4: In the section Treatment in the metastatic setting (5.2.) information on liver-directed treatment options is lacking (e.g. RFA, TACE, SIRT).

We thank the reviewer for his suggestion. We have added a new subsection on “liver-directed therapies” in our treatment section.

Modification in the revised manuscript, Treatment section, page 8, line 325:

“5.2.2. Liver-directed therapies

Interventional radiology plays an iportant role in the therapeutic management of well-differentiated low-grade NEN, especially when liver burden is important or in the presence of a secretory syndrome [81–83]. Various types of liver-directed therapies are currently available, including intra-arterial therapies such as transarterial embolization (TAE), transarterial chemoembolization (TACE) and selective internal radiation therapy (SIRT), or radiofrequency ablation (RFA). In a small series of metastatic GEP-NEN G-3, one retrospective study suggested that aggressive locoregional treatment, including treatment with intra-arterial liver-directed therapies, improved OS compared with administration of systemic chemotherapy alone [84]. To date, there is not enough data to recommend this therapeutic approach in routine practice for NEC.”
